# Automatic Domain Adaptation by Transformers in In-Context Learning

**Ryuichiro Hataya** [1]  **Kota Matsui** [2 1]  **Masaaki Imaizumi** [3 1]

## Abstract

Selecting or designing an appropriate domain adaptation algorithm for a given problem remains challenging. This paper presents a Transformer model that can provably approximate and opt for unsupervised domain adaptation (UDA) methods for a given dataset in the in-context learning framework, where a foundation model performs new tasks without updating its parameters at test time. Specifically, we prove that i) Transformers can approximate instance-based and feature-based unsupervised domain adaptation algorithms, and ii) automatically select the approximated algorithms suited for a given dataset. Numerical results indicate that in-context learning demonstrates an adaptive domain adaptation surpassing existing methods.

## 1. Introduction

Domain adaptation provides a methodology for transferring "knowledge" obtained in one domain to another related domain (Ben-David et al., 2010). One of the challenges associated with domain adaptation is selecting and differentiating effective methods. There are many existing approaches in domain adaptation, such as the instance-based methods (Dai et al., 2007; Sugiyama et al., 2007; Kanamori et al., 2009) and the feature-based methods(Daumé III, 2009; Ganin et al., 2016). However, when each method is effective differs based on data. Specifically, the instance-based methods (Sugiyama et al., 2007; Kanamori et al., 2009) are effective when there exists a valid density ratio between covariates of the source and target domains, whilst the feature-based methods are effective if we can find a domain-invariant representation on a common feature space between the domains. To properly select an appropriate approach to the given data, it is essential to assess whether they meet the specific conditions of each method.

---
*Equal contribution [1]RIKEN AIP, Tokyo, Japan [2]Nagoya University, Aichi, Japan [3]The University of Tokyo, Tokyo, Japan. Correspondence to: Ryuichiro Hataya <ryuichiro.hataya@riken.jp>.

*Proceedings of the 1st Workshop on In-Context Learning at the 41st International Conference on Machine Learning*, Vienna, Austria. 2024. Copyright 2024 by the author(s).

As a method for developing versatile algorithms, the in-context learning ability of foundation models has garnered significant attention. In-context learning, a form of meta-learning in foundational models, allows models to adapt to new tasks without updating their parameters. In particular, the capabilities of Transformers in in-context learning have been intensively surveyed: for example, (Bai et al., 2023) showed that Transformers can learn algorithms such as gradient descent, as well as their optimal selection methods. Such in-context learning abilities of Transformers have been validated from both experimental and theoretical aspects (Garg et al., 2022; Li et al., 2023; Von Oswald et al., 2023; Akyürek et al., 2022; Xie et al., 2021; Zhang et al., 2023; Bai et al., 2023; Lin et al., 2023; Ahn et al., 2024; Raventós et al., 2024).

In this study, we demonstrate that Transformer models effectively address the challenge in in-context learning. Specifically, our analysis reveals that Transformers in the framework i) solve domain adaptation problems by approximating the main UDA algorithms, and ii) automatically select suitable methods adaptively to the dataset's characteristics. The results indicate that Transformers can, in context, not only implement various transfer learning methods but also possess the capability to appropriately select among them. These findings suggest that by accurately choosing a combination of these methods based on the dataset, performance can be enhanced beyond what is achievable by applying the methods individually. The full version of this manuscript can be found at https://arxiv.org/abs/2405.16819.

## 2. Preliminary

### 2.1. Unsupervised Domain Adaptation

**Setup** We consider the *Unsupervised domain adaptation* problem with two domains. Let $\mathcal{X} \subset \mathbb{R}^d$ be a compact input space and $\mathcal{Y}$ be an output space. Then, the source and the target domains, $P_S$ and $P_T$, are distributions on $\mathcal{X} \times \mathcal{Y}$, let their density functions be $p_S$ and $p_T$, and their marginal distributions on $\mathcal{X}$ be $P_S^X$ and $P_T^X$. Denote $\mathcal{D}_S = \{(\boldsymbol{x}_i^S, y_i^S)\}_{i=1}^n$ and $\mathcal{D}_T = \{\boldsymbol{x}_i^T\}_{i=1}^{n'}$ for the source labeled data and target unlabeled data, respectively and let $N := n + n'$. Given a loss function $\ell : \mathcal{Y} \times \mathcal{Y} \to \mathbb{R}_{\geq 0}$ and a hypothesis space $\mathcal{F}$, UDA aims to minimize the target risk $\operatorname{argmin}_{f \in \mathcal{F}} R_T(f)$,

where $R_T(f) \coloneqq \mathbb{E}_{(\boldsymbol{x},y) \sim P_T}[\ell(f(\boldsymbol{x}), y)]$ is the target risk, without observing labels of the data from the target domain. To this end, UDA methods aim to match the source and target distributions in the feature space, categorized into instance-based and feature-based approaches.

**Instance-based Methods** Instance-based approaches reweight instances from the source domain to minimize the target risk, based on the following transferability assumption called *covariate shift*.

**Assumption 1** (Covariate Shift, (Shimodaira, 2000)). *Suppose that the distribution of $y$ conditioned on the input $\boldsymbol{x}$ is the same in the source and target domains, i.e., $p_S(y|\boldsymbol{x}) = p_T(y|\boldsymbol{x})$.*

Under this assumption, $R_T(f) = \mathbb{E}_{(\boldsymbol{x},y) \sim P_S}(q(\boldsymbol{x})\ell(f(\boldsymbol{x}), y))$ holds, where $q(\boldsymbol{x}) = p_T(\boldsymbol{x})/p_S(\boldsymbol{x})$ is the density ratio between the marginal distributions of source and target. Then, *Importance-weighted learning (IWL)* (Sugiyama et al., 2012; Kimura & Hino, 2024) obtains $\operatorname{argmin}_{f \in \mathcal{F}} R_T(f)$ in two steps: 1. Empirically learn an estimator of the density ratio $\widehat{q}$ using $\{\boldsymbol{x}_i^S\}_{i=1}^n$ and $\mathcal{D}_T$, and then, 2. minimize the weighted empirical risk $\widehat{R}_T(f) = \frac{1}{n}\sum_{i=1}^n \widehat{q}(\boldsymbol{x}_i^S)\ell(f(\boldsymbol{x}_i^S), y_i^S)$ consists of data from the source domain.

Here, we introduce *unconstrained Least-Squares Importance Fitting (uLSIF)* (Kanamori et al., 2009), a method for density-ratio estimation. Firstly, the density ratio is estimated by the linear basis function model $\widehat{q}_{\boldsymbol{\alpha}}(\boldsymbol{x}) = \boldsymbol{\alpha}^\top \boldsymbol{\phi}(\boldsymbol{x}) = \sum_{j=1}^J \alpha_j \phi_j(\boldsymbol{x})$, where $\boldsymbol{\alpha} = [\alpha_1, \ldots, \alpha_J]$ and $\boldsymbol{\phi} = [\phi_1, \ldots, \phi_J]$ are parameter and feature vectors respectively, where $\phi_j : \mathcal{X} \mapsto \mathbb{R}$ are feature maps for an input $\boldsymbol{x}$. uLSIF estimates the weight $\boldsymbol{\alpha}$ to directly minimize the following squared error: $L(\boldsymbol{\alpha}) = \frac{1}{2}\int_{\mathcal{X}}(\widehat{q}_{\boldsymbol{\alpha}}(\boldsymbol{x}) - q(\boldsymbol{x}))^2 p_S(\boldsymbol{x})\mathrm{d}\boldsymbol{x}$. By some calculation, a minimizer of (2.1) is estimated $\widehat{\boldsymbol{\alpha}} = \operatorname{argmin}_{\boldsymbol{\alpha} \geq 0} \widehat{L}(\boldsymbol{\alpha})$ with

$$\widehat{L}(\boldsymbol{\alpha}) = \frac{1}{2n}\sum_{i=1}^n (\widehat{q}_{\boldsymbol{\alpha}}(\boldsymbol{x}))^2 - \frac{1}{n'}\sum_{i=1}^{n'} \widehat{q}_{\boldsymbol{\alpha}}(\boldsymbol{x}) + \frac{\lambda}{2}\|\boldsymbol{\alpha}\|_2^2 \quad (1)$$

where $\lambda > 0$ is a regularization parameter, $\Psi_{jj'} \coloneqq \frac{1}{n}\sum_{i=1}^n \phi_j(\boldsymbol{x}_i^S)\phi_{j'}(\boldsymbol{x}_i^S)$ and $\psi_j \coloneqq \frac{1}{n'}\sum_{i=1}^{n'}\phi_j(\boldsymbol{x}_i^T)$. Then, if $\mathcal{F}$ is a space of linear model, we can obtain a classifier with an empirical importance-weighted problem $\boldsymbol{w}^* = \operatorname{argmin}_{\boldsymbol{w} \in \mathbb{R}^J} \widehat{R}_T(\boldsymbol{w})$ with $\widehat{R}_T(\boldsymbol{w}) \coloneqq \sum_{(\boldsymbol{x},y) \in \mathcal{D}_S} \widehat{q}_{\widehat{\boldsymbol{\alpha}}}(\boldsymbol{x})\ell(\boldsymbol{w}^\top \boldsymbol{\phi}(\boldsymbol{x}), y)$. Then, we obtain a classifier defined as $\widehat{f}^{\text{IWL}}(\boldsymbol{x}) \coloneqq (\boldsymbol{w}^*)^\top \boldsymbol{\phi}(\boldsymbol{x})$.

**Feature-based Methods** A feature-based method learns a domain-invariant feature map $\phi : \mathcal{X} \to \mathcal{X}'$, where $\mathcal{X}'$ is a feature space, so that $R_T(f') \approx \mathbb{E}_{(\boldsymbol{x},y) \sim P_S}\ell(f'(\phi(\boldsymbol{x})), y)$, where $f' : \mathcal{X}' \to \mathcal{Y}$. The *domain adversarial neural network (DANN)* (Ganin et al., 2016) is a typical method to

achieve this by using adversarial learning. DANN consists of three modules, a feature extractor $f_F : \mathcal{X} \to \mathcal{X}'$, a label classifier $f_L : \mathcal{X}' \to [0, 1]$, and a domain discriminator $f_D : \mathcal{X}' \to [0, 1]$, parameterized by $\boldsymbol{\theta}_F, \boldsymbol{\theta}_L, \boldsymbol{\theta}_D$, respectively. Then, a classifier $f_L$ with an invariant feature extractor $f_F$ can be obtained by solving the following minimax problem:

$$\min_{\boldsymbol{\theta}_F, \boldsymbol{\theta}_L} \max_{\boldsymbol{\theta}_D} L(\boldsymbol{\theta}_F, \boldsymbol{\theta}_L) - \lambda \Omega(\boldsymbol{\theta}_F, \boldsymbol{\theta}_D). \quad (2)$$

where $\lambda > 0$ is a regularization parameter, $L(\boldsymbol{\theta}_F, \boldsymbol{\theta}_L) \coloneqq \sum_{(\boldsymbol{x},y) \in \mathcal{D}_S} \gamma(f_L \circ f_F(\boldsymbol{x}), y)$ is label classification loss and $\Omega(\boldsymbol{\theta}_F, \boldsymbol{\theta}_D) \coloneqq \frac{1}{n}\sum_{(\boldsymbol{x},y) \in \mathcal{D}_S} \gamma(f_D \circ f_F(\boldsymbol{x}), 0) + \frac{1}{n'}\sum_{(\boldsymbol{x},y) \in \mathcal{D}_T} \gamma(f_D \circ f_F(\boldsymbol{x}), 1)$ is domain classification loss. Here, $\gamma(p, q) \coloneqq -q\log p - (1 - q)\log(1 - p)$ for $p, q \in [0, 1]$ is a sigmoid cross entropy function. The model parameters are updated by gradient descent with a learning rate of $\eta$ as follows:

$$\boldsymbol{\theta}_F \leftarrow \boldsymbol{\theta}_F - \eta \nabla_{\boldsymbol{\theta}_F}\left(L(\boldsymbol{\theta}_F, \boldsymbol{\theta}_L) - \lambda \Omega(\boldsymbol{\theta}_F, \boldsymbol{\theta}_D)\right), \quad (3)$$

$$\boldsymbol{\theta}_L \leftarrow \boldsymbol{\theta}_L - \eta \nabla_{\boldsymbol{\theta}_L} L(\boldsymbol{\theta}_F, \boldsymbol{\theta}_L), \quad (4)$$

$$\boldsymbol{\theta}_D \leftarrow \boldsymbol{\theta}_D - \eta \lambda \nabla_{\boldsymbol{\theta}_D} \Omega(\boldsymbol{\theta}_F, \boldsymbol{\theta}_D). \quad (5)$$

### 2.2. In-context Learning

**Setup.** In in-context learning, a fixed Transformer observes a dataset $\mathcal{D} = \{(\boldsymbol{x}_i, y_i)\}_{i=1}^N \sim (P)^N$ with pairs of an input $\boldsymbol{x}_i$ and its label $y_i$ from a joint distribution $P$ and a new query input $\boldsymbol{x}_*$ then predicts $y_*$ corresponding to $\boldsymbol{x}_*$. Different from the standard supervised learning, in in-context learning, the Transformer is pre-trained on other datasets $\mathcal{D}'$ from different distributions to learn an algorithm to predict $y_*$. Unlike the standard meta learning, at the inference time of in-context learning, the parameters of a Transformer are fixed. Our interest is to study the expressive power of a Transformer model for algorithms on a given dataset $\mathcal{D}$.

**Transformer.** Define an $L$-layer Transformer consisting of $L$ Transformer layers as follows. The $l$th Transformer layer maps an input matrix $\boldsymbol{H}^{(l)} \in \mathbb{R}^{D \times N}$ to $\widetilde{\boldsymbol{H}}^{(l)} \in \mathbb{R}^{D \times N}$ and is composed of a self-attention block and a feed-forward block. The self-attention block $\text{Attn}^{(l)} : \mathbb{R}^{D \times N} \to \mathbb{R}^{D \times N}$ is parameterized by $D \times D$ matrices $\{(\boldsymbol{K}_m^{(l)}, \boldsymbol{Q}_m^{(l)}, \boldsymbol{V}_m^{(l)})\}_{m=1}^M$, where $M$ is the number of heads, and defined as $\text{Attn}^{(l)}(\boldsymbol{X}) = \boldsymbol{X} + \frac{1}{N}\sum_{m=1}^M \boldsymbol{V}_m^{(l)}\boldsymbol{X}\sigma((\boldsymbol{Q}_m^{(l)}\boldsymbol{X})^\top \boldsymbol{K}_m^{(l)}\boldsymbol{X})$. $\sigma$ denotes an activation function applied elementwisely.

The feed-forward block $\text{MLP}^{(l)} : \mathbb{R}^{D \times N} \to \mathbb{R}^{D \times N}$ is a multi-layer perceptron with a skip connection, parameterized by $(\boldsymbol{W}_1^{(l)}, \boldsymbol{W}_2^{(l)}) \in \mathbb{R}^{D' \times D} \times \mathbb{R}^{D \times D'}$, such that $\text{MLP}^{(l)}(\boldsymbol{X}) = \boldsymbol{X} + \boldsymbol{W}_2^{(l)}\varsigma(\boldsymbol{W}_1^{(l)}\boldsymbol{X})$, where $\varsigma$ is an activation function applied elementwisely. In this paper, we let

both $\sigma$ and $\varsigma$ be the ReLU function in this paper, following (Bai et al., 2023).

An $L$-layer Transformer $\mathrm{TF}_{\boldsymbol{\theta}}$, parameterized by $\boldsymbol{\theta} = \{(\boldsymbol{K}_1^{(l)}, \boldsymbol{Q}_1^{(l)}, \boldsymbol{V}_1^{(l)}, \ldots, \boldsymbol{K}_M^{(l)}, \boldsymbol{Q}_M^{(l)}, \boldsymbol{V}_M^{(l)}, \boldsymbol{W}_1^{(l)}, \boldsymbol{W}_2^{(l)})\}_{l=1}^L$, is a composition of the abovementioned layers as $\mathrm{TF}_{\boldsymbol{\theta}}(\boldsymbol{X}) = \mathrm{MLP}^{(L)} \circ \mathrm{Attn}^{(L)} \circ \cdots \circ \mathrm{MLP}^{(1)} \circ \mathrm{Attn}^{(1)}(\boldsymbol{X})$.

In the remaining text, the superscript to indicate the layer numer $^{(l)}$ is sometimes omitted for brevity, and the $n$th columns of $\boldsymbol{H}^{(l)}, \widetilde{\boldsymbol{H}}^{(l)}$ are denoted as $\boldsymbol{h}_n^{(l)}, \widetilde{\boldsymbol{h}}_n^{(l)}$. We use $\|\boldsymbol{\theta}\|_{\mathrm{TF}}$ as a norm of the parameter matrices; see Section A in the appendix.

# 3. Approximating UDA Algorithms by Transformers

This section demonstrates that transformers in in-context learning can approximate existing UDA algorithms. Specifically, we show that transformers can approximate uLSIF-based IWL, an instance-based method, and DANN, a feature-based method.

## 3.1. Setup and Notion

We consider in-context domain adaptation, where a fixed Transformer model is given a tuple $(\mathcal{D}_S, \mathcal{D}_T, \boldsymbol{x}_*)$, where $\boldsymbol{x}_* \sim p_T^X(\boldsymbol{x})$, and predicts $y_*$ without updating model parameters.

We construct the input matrix using both source and target data. Specifically, we suppose the input data, namely $(\boldsymbol{x}_i^S, y_i^S) \in \mathcal{D}_S$ and $\boldsymbol{x}_i^T \in \mathcal{D}_T$, are encoded into $\boldsymbol{H}^{(1)} \in \mathbb{R}^{D \times (N+1)}$ as follows:

$$\boldsymbol{H}^{(1)} = \begin{bmatrix} \boldsymbol{x}_1 & \ldots & \boldsymbol{x}_n & \boldsymbol{x}_{n+1} & \ldots & \boldsymbol{x}_N & \boldsymbol{x}_{N+1} \\ y_1 & \ldots & y_n & 0 & \ldots & 0 & 0 \\ t_1 & \ldots & t_n & t_{n+1} & \ldots & t_N & t_{N+1} \\ s_1 & \ldots & s_n & s_{n+1} & \ldots & s_N & s_{N+1} \\ 1 & \ldots & 1 & 1 & \ldots & 1 & 1 \\ \boldsymbol{0}_{D-(d+4)} & \ldots & \boldsymbol{0}_{D-(d+4)} & \boldsymbol{0}_{D-(d+4)} & \ldots & \boldsymbol{0}_{D-(d+4)} & \boldsymbol{0}_{D-(d+4)} \end{bmatrix}, \tag{6}$$

where $\boldsymbol{x}_i = \boldsymbol{x}_i^S$ and $y_i = y_i^S$ for $1 \leq i \leq n$, $\boldsymbol{x}_i = \boldsymbol{x}_{i-n}^T$ for $n+1 \leq i \leq N$, $\boldsymbol{x}_{N+1} = \boldsymbol{x}_*$. $t_i = 1$ for $1 \leq i \leq n$ otherwise 0 is used to indicate which data point is from the source domain, and $s_i = 1$ for $1 \leq i \leq N$ otherwise 0 is used to mark training data. We further define an output of a transformer for the prediction corresponding to $\boldsymbol{x}_{N+1} = \boldsymbol{x}_*$. For an input $\boldsymbol{H}^{(1)}$ and its corresponding output matrix $\mathrm{TF}_{\boldsymbol{\theta}}(\boldsymbol{H}^{(1)})$, we write its $(2, N+1)$-th element of $\mathrm{TF}_{\boldsymbol{\theta}}(\boldsymbol{H}^{(1)})$ as $\mathrm{TF}_{\boldsymbol{\theta}}^*(\boldsymbol{H}^{(1)})$.

Let $B_x, B_y > 0$. To use $(\epsilon, R, M, C)$-approximability in Appendix A, we suppose $\|\boldsymbol{x}_i\| \leq B_x$ and $\|y_i\| \leq B_y$ for any $i \in \{1, \ldots, N+1\}$, and $\|\boldsymbol{w}\| \leq B_w$ for a model weight $\boldsymbol{w}$ in in-context learning. Additionally, for the feature map of IWL, we suppose $\|\boldsymbol{\phi}\| \leq 1$ so that $\|\boldsymbol{\phi}(\boldsymbol{x})\| \leq B_x$.

## 3.2. In-context IWL with uLSIF

We show a transformer in the ICL scheme can approximate the uLSIF estimator-based IWL algorithm, which we refer to as IWL in the following.

In this section, we consider the following setup: Fix any $B_w > 0$, $B_\alpha > 0$, $L_1, L_2 > 0$, $\eta_1$, and $\eta_2$. Given a loss function $\ell$ that is convex in the first argument, and $\nabla_1 \ell$ is $(\epsilon, R, M, C)$-approximable by the sum of ReLUs with $R = \max(B_w B_x, B_y, 1)$. We first state the main result of this section. This theorem shows that the transformer has the performance to function as an algorithm almost equivalent to IWL for any input $\boldsymbol{x}_*$. Note that the query $\boldsymbol{x}_*$ is encoded in the input matrix $\boldsymbol{H}^{(1)}$ defined in (6).

**Theorem 1.** *Consider* $\widehat{f}^{\mathrm{IWL}}$ *with* $\boldsymbol{\phi}$ *which is approximable by a sum of ReLUs. Fix* $\varepsilon > 0$ *arbitrarily and set* $L_2$ *as satisfying* $0 < \epsilon \leq B_w/2L_2$. *Suppose that an input* $(\mathcal{D}_S, \mathcal{D}_T, \boldsymbol{x}_*)$ *satisfies that* $\sup_{\boldsymbol{w}: \|\boldsymbol{w}\|_2 \leq B_w} \lambda_{\max}(\nabla^2 \widehat{R}_T(\boldsymbol{w}; \mathcal{D}_S)) \leq \eta_2/2$ *and the minimizer* $\boldsymbol{w}^* \in \arg\min \widehat{R}_T(\boldsymbol{w}; \mathcal{D}_S)$ *satisfies* $\|\boldsymbol{w}^*\| \leq B_w/2$. *Then, there exists a transformer* $\mathrm{TF}_{\boldsymbol{\theta}}$ *with* $L_1 + L_2 + 1$ *layers and* $M$ *heads which satisfies the following:* $\|\mathrm{TF}_{\boldsymbol{\theta}}^*(\boldsymbol{H}^{(1)}) - \widehat{f}^{\mathrm{IWL}}(\boldsymbol{x}_*)\| \leq \varepsilon$.

This result has several implications. First, while transformers with ICL have been shown to approximate algorithms for i.i.d. data (Bai et al., 2023), this theorem extends it to show the capability of handling domain shifts. This extension is nontrivial since it requires showing that the Transformer can express the algorithm's ability to correct the domain shifts. Second, this theorem shows that Transformers do not need to specify the feature map $\boldsymbol{\phi}(\cdot)$. Specifically, whatever feature map $\boldsymbol{\phi}(\cdot)$ is used for uLSIF, the transformer can approximate the uLSIF based on it. In other words, pre-training a transformer induces a situation-specific feature map $\boldsymbol{\phi}(\cdot)$.

## 3.3. In-context DANN

We show the existence of a Transformer that can approximate DANN, presented in Section 2.1.

At the beginning, we identify the structure of the DANN to be approximated. First, we assume that a label classifier $f_L \circ f_F(\boldsymbol{x})$ is represented by a two-layer neural network model $\Lambda(\boldsymbol{x}; \boldsymbol{U}, \boldsymbol{w}) := \sum_{k=1}^K w_k r(\boldsymbol{u}_k^\top \boldsymbol{x})$, and the domain classifier $f_D \circ f_F(\boldsymbol{x})$ also follows a similar model $\Delta(\boldsymbol{x}; \boldsymbol{U}, \boldsymbol{v}) := \sum_{k=1}^K v_k r(\boldsymbol{u}_k^\top \boldsymbol{x})$, with parameters $\boldsymbol{w}, \boldsymbol{v} \in \mathbb{R}^K$ and $\boldsymbol{U} = [\boldsymbol{u}_1, \ldots \boldsymbol{u}_K] \in \mathbb{R}^{K \times d}$ and an activation function $r(\cdot)$. Let $\boldsymbol{u} = \mathrm{vec}(\boldsymbol{U})$ and $\boldsymbol{w}, \boldsymbol{v}, \boldsymbol{u}$ correspond to $\boldsymbol{\theta}_L, \boldsymbol{\theta}_D, \boldsymbol{\theta}_F$, respectively. This simplification using a two-layer neural network was included to simplify the discussion and is easy to generalize.

We further define a sequence of parameters

$\{(\boldsymbol{u}^{(l)}, \boldsymbol{v}^{(l)}, \boldsymbol{w}^{(l)})\}_{l=1}^L$ that can be obtained by $L$ updates by the DANN. In preparation, we define a closed set $\mathcal{W} = \{(\boldsymbol{w}, \boldsymbol{v}, \boldsymbol{u}) : \|\boldsymbol{w}\| \leq B_w, \|\boldsymbol{v}\| \leq B_v, \max_k \|\boldsymbol{u}_k\| \leq B_u\} \subset \mathbb{R}^{K \times K \times Kd}$ with some $B_u, B_w, B_v > 0$ and also define a projection $\Pi_{\mathcal{W}}(\cdot)$ onto $\mathcal{W}$. Let $(\boldsymbol{w}^{(1)}, \boldsymbol{v}^{(1)}, \boldsymbol{u}^{(1)}) \in \mathcal{W}$ be a tuple of initial values. Then, the subsequent parameters are recursively defined by the following updates:

$$\boldsymbol{u}^{(l)} = \Pi_{\mathcal{W}}\left(\boldsymbol{u}^{(l-1)} - \eta\nabla_{\boldsymbol{u}}\{L(\boldsymbol{u}^{(l-1)}, \boldsymbol{w}^{(l-1)})\right.$$
$$\left. -\lambda\Omega(\boldsymbol{u}^{(l-1)}, \boldsymbol{v}^{(l-1)})\}\right), \quad (7)$$

$$\boldsymbol{w}^{(l)} = \Pi_{\mathcal{W}}\left(\boldsymbol{w}^{(l-1)} - \eta\nabla_{\boldsymbol{w}}L(\boldsymbol{u}^{(l-1)}, \boldsymbol{w}^{(l-1)})\right), \quad (8)$$

$$\boldsymbol{v}^{(l)} = \Pi_{\mathcal{W}}\left(\boldsymbol{v}^{(l-1)} - \eta\lambda\nabla_{\boldsymbol{v}}\Omega(\boldsymbol{u}^{(l-1)}, \boldsymbol{v}^{(l-1)})\right). \quad (9)$$

These updates are an analogy of the original DANN updates Equations (3) to (5) to optimize the minimax loss function Equation (2) of DANN depending on $\mathcal{D}_S$ and $\mathcal{D}_T$. Here, we introduce the projection to explain a more realistic setup, for example, (Shen et al., 2018). Finally, the classifier obtained by $L$ updates of DANN is defined as $\widehat{f}^{\mathrm{DANN}}(\boldsymbol{x}) := \Lambda(\boldsymbol{x}; \boldsymbol{u}^{(L)}, \boldsymbol{w}^{(L)})$.

We show the existence of a Transformer that successively approximates the parameter sequence by DANN defined above. The results are given in the following statement.

**Theorem 2.** *Fix any $B_u, B_w, B_v > 0$, $L > 0$, $\eta > 0$, and $\epsilon > 0$. Suppose activation function $r$ is $C^4$-smooth. Then, there exists a $2L$-layer Transformer $\mathrm{TF}_{\boldsymbol{\theta}}$ with $\max_{l \in \{1,\ldots,2L\}} M^{(l)} \leq \widetilde{\mathcal{O}}(\epsilon^{-2})$, $\max_{l \in \{1,\ldots,2L\}} D^{(l)} \leq \widetilde{\mathcal{O}}(\epsilon^{-2}) + D_{\mathrm{MLP}}$, $\|\boldsymbol{\theta}\|_{\mathrm{TF}} \leq \mathcal{O}(1+\eta) + C_{\mathrm{MLP}}$, with some existing constants $D_{\mathrm{MLP}}, C_{\mathrm{MLP}} > 0$, which satisfies the following: for any input $(\mathcal{D}_S, \mathcal{D}_T, \boldsymbol{x}_*)$, an $2l$-th layer of the transformer maps $\widetilde{\boldsymbol{h}}_i^{(2l-1)} := [\boldsymbol{z}_i, \widetilde{\boldsymbol{u}}^{(l-1)}, \widetilde{\boldsymbol{w}}^{(l-1)}, \widetilde{\boldsymbol{v}}^{(l-1)}]$ with any $(\widetilde{\boldsymbol{w}}^{(l-1)}, \widetilde{\boldsymbol{v}}^{(l-1)}, \widetilde{\boldsymbol{u}}^{(l-1)}) \in \mathcal{W}$ to $\widetilde{\boldsymbol{h}}_i^{(2l)} := [\boldsymbol{z}_i, \widetilde{\boldsymbol{u}}^{(l)}, \widetilde{\boldsymbol{w}}^{(l)}, \widetilde{\boldsymbol{v}}^{(l)}]$ for each $i \in \{1,\ldots,N+1\}$ to satisfy Equations (7) to (9) where $\boldsymbol{\epsilon}_u^{(l-1)}, \boldsymbol{\epsilon}_w^{(l-1)}, \boldsymbol{\epsilon}_v^{(l-1)} \in \mathbb{R}^k$ are some vectors satisfying $\max\{\|\boldsymbol{\epsilon}_u^{(l-1)}\|, \|\boldsymbol{\epsilon}_w^{(l-1)}\|, \|\boldsymbol{\epsilon}_v^{(l-1)}\|\} \leq \epsilon$.*

The results show that the existing Transformer approximates the original DANN updates Equations (7) to (9), at each step. The terms $\boldsymbol{\epsilon}_u^{(l-1)}, \boldsymbol{\epsilon}_w^{(l-1)}, \boldsymbol{\epsilon}_v^{(l-1)}$ expresses approximation errors, whose norm are no more than the fixed $\epsilon$. Consequently, the final output of the transformer approximates the output of the original DANN after $L$-iterations. The results are given as follows:

**Corollary 1.** *Consider the setup as in Theorem 2. Then, for any input $(\mathcal{D}_S, \mathcal{D}_T, \boldsymbol{x}_*)$, the transformer $\mathrm{TF}_{\boldsymbol{\theta}}$ in Theorem 2 outputs a corresponding tuple $(\widetilde{\boldsymbol{w}}^{(L)}, \widetilde{\boldsymbol{v}}^{(L)}, \widetilde{\boldsymbol{u}}^{(L)}) \in \mathcal{W}$ which satisfies $\|\mathrm{TF}_{\boldsymbol{\theta}}^*(\boldsymbol{H}^1) - \widehat{f}^{\mathrm{DANN}}(\boldsymbol{x}_*)\| \leq L\varepsilon$.*

# 4. Automatic Algorithm Selection by In-Context UDA

We demonstrate that Transformers can automatically select UDA algorithms based on data in context. Specifically, we consider the selection of methods determined by whether the supports of the source distribution $p_S$ and the target distribution $p_T$ sufficiently overlap. In cases where the support of the target distribution $p_T$ overlaps with that of the source distribution $p_S$, the IWL algorithm, which operates using the density ratio $q(\boldsymbol{x}) = p_T(\boldsymbol{x})/p_S(\boldsymbol{x})$, is employed. In other words, IWL should be employed when we have

$$p_S(\boldsymbol{x}) > 0 \text{ holds for all } \boldsymbol{x} \text{ such that } p_T(\boldsymbol{x}) > 0. \quad (10)$$

Conversely, when there is no such overlap, i.e., $p_S(\boldsymbol{x}) = 0$ for some $\boldsymbol{x}$ with $p_T(\boldsymbol{x}) > 0$, one should select DANN, which does not rely on the density ratio.

We show that Transformers are capable of realizing the above design automatically.

$$\widehat{f}^{\mathrm{ICUDA}}(\boldsymbol{x}) := \begin{cases} \widehat{f}^{\mathrm{IWL}}(\boldsymbol{x}_*) & \text{if } \min_{\boldsymbol{x}:p_T(\boldsymbol{x})>0} p_S(\boldsymbol{x}) > 0, \\ \widehat{f}^{\mathrm{DANN}}(\boldsymbol{x}_*) & \text{otherwise,} \end{cases}$$
$$(11)$$

The statement is as follows:

**Theorem 3.** *Fix any $\epsilon > 0$. Suppose that $n' \geq (1/\epsilon)^3 \log n'$ holds and $p_T(\cdot)$ is Lipschitz continuous. Then, there exists a Transformer $\mathrm{TF}_{\boldsymbol{\theta}}$ with three layers and $M$ heads satisfies the following with probability at least $1 - 1/n' - O(\epsilon)$: for any input $(\mathcal{D}_S, \mathcal{D}_T, \boldsymbol{x}_*)$, a transformer $\mathrm{TF}_{\boldsymbol{\theta}}$ satisfies*

$$\|\mathrm{TF}_{\boldsymbol{\theta}}^*(\boldsymbol{H}^1) - \widehat{f}^{\mathrm{ICUDA}}(\boldsymbol{x}_*)\| \leq \epsilon. \quad (12)$$

The results show that the Transformer can automatically check the condition (10) and use the appropriate algorithm without the user having to make a choice. Note that the density ratio condition (10) is one of the options that are automatically learned, and in practice, Transformers can learn more complicated conditions. Importantly, in this case, cross-validation cannot be used to make a selection since true labels of the target domain data cannot be observed.

# 5. Conclusion and Discussion

This paper proved that Transformers can approximate instance-based and feature-based domain adaptation algorithms and select an appropriate one for the given dataset in the in-context learning framework. Technically, our results revealed that Transformers can approximately solve some types of linear equations and minimax problems. Numerical experiments in Figure 1 demonstrated that domain adaptation in in-context learning outperforms UDA methods tailored for specific assumptions.

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

## A. Additional Setup

We define a norm of a parameter $\|\boldsymbol{\theta}\|_{\mathrm{TF}}$ of Transformers as

$$\|\boldsymbol{\theta}\|_{\mathrm{TF}} = \max_{l\in\{1,\dots,L\}} \left\{ \max_{m\in\{1,\dots,M\}} \left\{ \left\|\boldsymbol{Q}_m^{(l)}\right\|, \left\|\boldsymbol{K}_m^{(l)}\right\| \right\} + \sum_{m=1}^{M} \left\|\boldsymbol{V}_m^{(l)}\right\| + \left\|\boldsymbol{W}_1^{(l)}\right\| + \left\|\boldsymbol{W}_2^{(l)}\right\| \right\}, \tag{13}$$

where $\|\cdot\|$ for matrices indicates the operator norm in this equation.

We also define the concepts necessary for our theoretical results. To approximate smooth functions, such as loss functions $\ell$ and $\gamma$, we use the following notion.

**Definition 1** (($\varepsilon, R, M, C$)-approximability by sum of ReLUs, (Bai et al., 2023)). *For $\varepsilon > 0$ and $R \geq 1$, a function $g : \mathbb{R}^k \to \mathbb{R}$ is ($\epsilon, R, M, C$)-approximabile by sum of ReLUs if there exist a function $f(\boldsymbol{z}) = \sum_{m=1}^{M} c_m \sigma(\boldsymbol{a}_m^\top \boldsymbol{z} + b_m)$ with $\sum_{m=1}^{M} |c_m| \leq C$ with $\max_{m\in\{1,\dots,M\}} \|\boldsymbol{a}\|_1 + b_m \leq 1$, where $\boldsymbol{a}_m \in \mathbb{R}^k$, $b_m \in \mathbb{R}$, $c_m \in \mathbb{R}$,, and $\sup_{\boldsymbol{z}\in[-R,R]^k} |g(\boldsymbol{z}) - f(\boldsymbol{z})| \leq \varepsilon$.*

All functions approximated in this paper are included in this class.

## B. Proof Outline of In-context IWL

In our proof, we construct three types of sub-transformers and approximate IWL-uLSIF by combining them. Specifically, the first sub-transformer with one layer to construct a feature map $\phi(\cdot)$, the second one calculates $\boldsymbol{\alpha}$ with $L_1$ layers, and this one optimizes the model weight $\boldsymbol{w} \in \mathbb{R}^J$ with $L_2$ layers. For simplicity, we define a notation $\boldsymbol{z}_i = [\boldsymbol{x}_i, y_i, t_i, s_i, 1]$. In the following, we describe each approximation step by step.

**Step 1: Feature map approximation.** First, we construct a transformer that approximates the feature map $\phi(\cdot)$. This construction is trivial under the assumption that the feature map $\phi(\cdot)$ is approximable by the sum of ReLUs. For example, a self-attention block can construct a feature map with the RBF kernel.

**Lemma 1.** *There exists a transformer with one layer which maps $\boldsymbol{h}_i^{(1)} = [\boldsymbol{z}_i, \boldsymbol{0}_J, \boldsymbol{0}_{D-(d+4)}]$ to $\widetilde{\boldsymbol{h}}_i^{(1)} = [\boldsymbol{z}_i, \widehat{\boldsymbol{\phi}}(\boldsymbol{x}_i), \boldsymbol{0}_{D-J-(d+4)}]$ for $i = 1, \dots, N+1$, such that $\|\boldsymbol{\phi} - \widehat{\boldsymbol{\phi}}\|_{L^\infty} \leq \varepsilon$ holds.*

**Step 2: Density ratio parameter approximation.** Second, we construct a transformer that approximates $\widehat{\boldsymbol{\alpha}}$ that minimizes the loss $\widehat{L}(\boldsymbol{\alpha})$ in Equation (1), in which a technical difficulty lies. For the approximation, we define a sequence of parameters $\{\boldsymbol{\alpha}_{\mathrm{GD}}^{(l)}\}_{l=1,2,\dots}$ that converge to $\widehat{\boldsymbol{\alpha}}$, using an update equation by the gradient descent algorithm:

$$\boldsymbol{\alpha}_{\mathrm{GD}}^{(l+1)} = \boldsymbol{\alpha}_{\mathrm{GD}}^{(l)} - \eta_1 \nabla\widehat{L}(\boldsymbol{\alpha}_{\mathrm{GD}}^{(l)}), \; l = 1, 2, \dots, \text{ and } \boldsymbol{\alpha}_{\mathrm{GD}}^{(1)} = \boldsymbol{0}_J. \tag{14}$$

Then, we develop a Transformer layer that exactly implements a single update step of this equation. We give the following statement.

**Lemma 2.** *There exists a Transformer with $L_1$ layer which maps $\widetilde{\boldsymbol{h}}_i^{(1)} = [\boldsymbol{z}_i, \boldsymbol{\phi}(\boldsymbol{x}_i), \boldsymbol{0}_J, \boldsymbol{0}_{D-2J-(d+4)}]$ to $\widetilde{\boldsymbol{h}}_i^{(L_1+1)} = [\boldsymbol{z}_i, \boldsymbol{\phi}(\boldsymbol{x}_i), \widetilde{\boldsymbol{\alpha}}^{(L_1)}, \boldsymbol{0}_{D-2J-(d+4)}]$ for $i = 1, \dots, N+1$, such that $\boldsymbol{\alpha}_{\mathrm{GD}}^{(L_1)} = \widetilde{\boldsymbol{\alpha}}^{(L_1)}$.*

We note that this approach with the gradient descent is more efficient in the sense of layer size than the direct minimization of $\widehat{L}(\boldsymbol{\alpha})$ using an inverse matrix.

**Step 3: Model parameter approximation.** In this step, we employ a similar approach to develop a Transformer that approximates $\boldsymbol{w}^*$, which is the minimizer of $\widehat{R}_T(\boldsymbol{w})$ in Section 2.1. To the aim, we define a sequence of parameters $\{\boldsymbol{w}_{\mathrm{GD}}^{(l)}\}_{l=1,2,\dots}$ that converge to $\boldsymbol{w}^*$ by the gradient descent algorithm:

$$\boldsymbol{w}_{\mathrm{GD}}^{(l+1)} = \boldsymbol{w}_{\mathrm{GD}}^{(l)} - \eta_2 \nabla\widehat{R}_T(\boldsymbol{w}_{\mathrm{GD}}^{(l)}), \; l = 1, 2, \dots, \text{ and } \boldsymbol{w}_{\mathrm{GD}}^{(1)} = \boldsymbol{0}_J. \tag{15}$$

Then, we develop a Transformer layer that approximates this algorithm as follows:

**Lemma 3.** *There exists a Transformer with $L_2$ layer which maps $\widetilde{\boldsymbol{h}}_i^{(L_1+1)} = [\boldsymbol{z}_i, \boldsymbol{\phi}(\boldsymbol{x}_i), \widetilde{\boldsymbol{\alpha}}^{(L_1)}, \boldsymbol{0}_{D-2J-(d+4)}]$ to $\widetilde{\boldsymbol{h}}_i^{(L_1+L_2+1)} = [\boldsymbol{z}_i, \boldsymbol{\phi}(\boldsymbol{x}_i), \widetilde{\boldsymbol{\alpha}}^{(L_1)}, \widetilde{\boldsymbol{w}}^{(L_2)}, 0]$ for $i = 1, \dots, N+1$, such that $\|\boldsymbol{w}_{\mathrm{GD}}^{(L_2)} - \widetilde{\boldsymbol{w}}^{(L_2)}\|_{L^\infty} \leq \varepsilon L_2 \eta_2 B_x$ holds.*

Combining these layers constitutes the transformer that achieves the approximation capability of Theorem 1. Importantly, this method requires as many layers as there are iterations, but this can be shortened by using regression coupling.

## C. Proof Outline of In-context DANN

Here, we focus on the sub-problem that a pair of Transformer layers approximates a single step of update of DANN consisting of two-layer neural networks. Specifically, the attention block at the first layer approximates the forward passes of networks, the succeeding MLP block computes the partial derivatives of the loss function, then the next attention block implements the minimax optimization step, and the last MLP block projects the weights.

The specific procedure is presented in the following steps. For brebity, we write $\boldsymbol{\gamma} := (\boldsymbol{u}, \boldsymbol{w}, \boldsymbol{v})$.

**Step 1: Forward pass approximation.** The first attention block performs the forward passes, that is, $\boldsymbol{x}_i \mapsto (\Lambda(\boldsymbol{x}_i), \Delta(\boldsymbol{x}_i))$, by approximating the activation function $r$ by sum-of-ReLUs.

> **Lemma 4.** *For each $\varepsilon > 0$, there exists an attention block which maps $\boldsymbol{h}_i = [\boldsymbol{z}_i, \boldsymbol{\gamma}, \boldsymbol{0}_4]$ to $\boldsymbol{h}_i' = [\boldsymbol{z}_i, \boldsymbol{\gamma}, \widetilde{\Lambda}(\boldsymbol{x}_i), \widetilde{\Delta}(\boldsymbol{x}_i), \boldsymbol{0}_2]$ for $i = 1, \ldots, N$ such that $\|\widetilde{\Lambda} - \Lambda\|_{L^\infty} \leq \varepsilon$ and $\|\widetilde{\Delta} - \Delta\|_{L^\infty} \leq \varepsilon$ hold.*

**Step 2: Loss derivative approximation.** The next MLP block obtains loss derivatives, i.e., $(\widetilde{\Lambda}(\boldsymbol{x}_i), \widetilde{\Delta}(\boldsymbol{x}_i), y_i, t_i, s_i) \mapsto (\mathbb{1}(i \leq n)\partial_1\gamma(\widetilde{\Lambda}(\boldsymbol{x}_i), y_i), \mathbb{1}(i \leq N)\partial_1\gamma(\widetilde{\Delta}(\boldsymbol{x}_i), y_i))$. We also use sum-of-ReLUs to approximate $\partial_1\gamma$.

> **Lemma 5.** *For any $\varepsilon > 0$, there exists an MLP block which maps $\boldsymbol{h}_i' = [\boldsymbol{z}_i, \boldsymbol{\gamma}, \widetilde{\Lambda}(\boldsymbol{x}_i), \widetilde{\Delta}(\boldsymbol{x}_i), \boldsymbol{0}_2]$ to $\boldsymbol{h}_i'' = [\boldsymbol{z}_i, \boldsymbol{\gamma}, \widetilde{\Lambda}(\boldsymbol{x}_i), \widetilde{\Delta}(\boldsymbol{x}_i), g_{\Lambda,i}, g_{\Delta,i}]$, where $g_{\Lambda,i}$ and $g_{\Delta,i}$ are scalars satisfying $|g_{\Lambda,i} - \mathbb{1}(i \leq n)\partial_1\gamma(\widetilde{\Lambda}(\boldsymbol{x}_i), y_i)| \leq \varepsilon$ and $|g_{\Delta,i} - \mathbb{1}(i \leq N)\partial_1\gamma(\widetilde{\Delta}(\boldsymbol{x}_i), y_i)| \leq \varepsilon$.*

**Step 3: Gradient descent approximation** The attention block at the second layer approximates the optimization step: $(\boldsymbol{u}, \boldsymbol{w}, \boldsymbol{v}) \mapsto (\boldsymbol{u} - \eta\nabla_{\boldsymbol{u}}(L(\boldsymbol{u}, \boldsymbol{w}) - \lambda\Omega(\boldsymbol{u}, \boldsymbol{v})), \boldsymbol{w} - \eta\nabla_{\boldsymbol{w}}L(\boldsymbol{u}, \boldsymbol{w}), \boldsymbol{v} - \eta\lambda\nabla_{\boldsymbol{v}}\Omega(\boldsymbol{u}, \boldsymbol{v}))$, which can be obtained by approximating $(s, t) \mapsto s \cdot r'(t)$.

> **Lemma 6.** *For each $\varepsilon > 0$, there exists an attention block that maps $\boldsymbol{h}_i'' = [\boldsymbol{z}_i, \boldsymbol{\gamma}, \widetilde{\Lambda}(\boldsymbol{x}_i), \widetilde{\Delta}(\boldsymbol{x}_i), g_{\Lambda,i}, g_{\Delta,i}]$ to $\boldsymbol{h}_i''' = [\boldsymbol{z}_i, \boldsymbol{u} - \eta\boldsymbol{g}_u, \boldsymbol{w} - \eta\boldsymbol{g}_w, \boldsymbol{v} - \eta\boldsymbol{g}_v, \widetilde{\Lambda}(\boldsymbol{x}_i), \widetilde{\Delta}(\boldsymbol{x}_i), g_{\Lambda,i}, g_{\Delta,i}]$, where $\|\boldsymbol{g}_u - \nabla_{\boldsymbol{u}}(\widetilde{L}(\boldsymbol{u}, \boldsymbol{w}) - \lambda\widetilde{\Omega}(\boldsymbol{u}, \boldsymbol{v}))\|_2, \|\boldsymbol{g}_w - \nabla_{\boldsymbol{w}}\widetilde{L}(\boldsymbol{u}, \boldsymbol{w})\|_2, \|\boldsymbol{g}_v - \lambda\nabla_{\boldsymbol{v}}\widetilde{\Omega}(\boldsymbol{u}, \boldsymbol{v})\|_2 \leq \varepsilon$.*

**Step 4: Projection approximation.** The last MLP block projects $\boldsymbol{u}, \boldsymbol{w}, \boldsymbol{v}$ onto $\mathcal{W}$ appropriately by the assumption.

By stacking such pairs of layers, a $2L$-layer Transformer can implement $L$ steps of DANN in context.

## D. Proof Outline of In-context Algorithm Selection

The key idea is to first implement kernel density estimation of $p_S(\boldsymbol{x})$ for $\boldsymbol{x} \in \mathcal{D}_T$ and then select an algorithm with $\min_{\boldsymbol{x} \in \mathcal{D}_T} \widehat{p}_S(\boldsymbol{x}) > \delta$, where $\delta > 0$ is a given threshold. Here, we assume that each token is $\boldsymbol{h}_i = [\boldsymbol{z}_i, \boldsymbol{\gamma}, \widetilde{f}^{\text{IWL}}(\boldsymbol{x}_i), \widetilde{f}^{\text{DANN}}(\boldsymbol{x}_i), \boldsymbol{0}]$, where $\boldsymbol{\gamma}$ consists of weights of in-context IWL and DANN and is unused in this selection process.

**Step 1: Approximation of density estimation.** The first step is to approximate the density function $p_S$ with kernel density estimation $\widehat{p}_S(\cdot) = \frac{1}{n}\sum_{i=1}^n K(\cdot, \boldsymbol{x}_i)$, where $K$ is a kernel, such as the RBF kernel, and evaluate it on each data point.

> **Lemma 7.** *For any $\epsilon > 0$, there exists an attention block that maps $\boldsymbol{h}_i$ to $\boldsymbol{h}_i' = [\boldsymbol{z}_i, \boldsymbol{\gamma}, \widetilde{f}^{\text{IWL}}(\boldsymbol{x}_i), \widetilde{f}^{\text{DANN}}(\boldsymbol{x}_i), p_i, \boldsymbol{0}]$, where $|p_i - \bar{p}_S(\boldsymbol{x}_i)| \leq \epsilon$.*

**Step 2: Algorithm selection approximation.** Then, the Transformer decides which algorithm is appropriate based on $\text{softmin}_{\beta,i} p_i > \delta$, where $\text{softmin}_{\beta,i} p_i = -\frac{1}{\beta}\log\sum(-\beta p_i)$ is a relaxation of $\min$ with an inverse temperature parameter $\beta > 0$.

> **Lemma 8.** *For any $\epsilon > 0$, there is an MLP block succeeded by a Transformer layer that maps $\boldsymbol{h}_{N+1}'$ to $\boldsymbol{h}_{N+1}'' = [\boldsymbol{z}_i, \boldsymbol{\gamma}, \widetilde{f}^{\text{IWL}}(\boldsymbol{x}_*), \widetilde{f}^{\text{DANN}}(\boldsymbol{x}_*), p_i, -\frac{1}{\beta}\log\sum_i \exp(-\beta p_i), \widetilde{f}(\boldsymbol{x}_*)]$, where $|\widetilde{f} - \widehat{f}^{\text{ICUDA}}| \leq \epsilon$.*

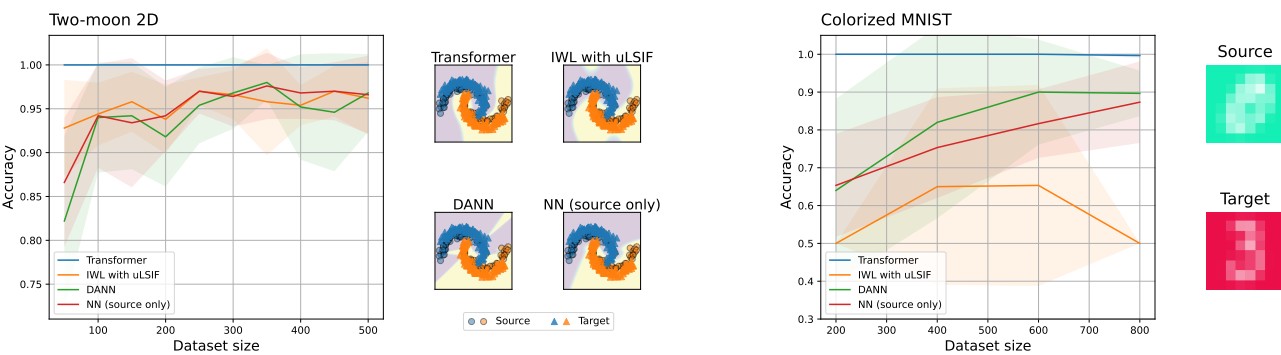

*Figure 1.* (Left) Test accuracy averaged over five runs of Transformer (ICL) and baseline models on the two-moon 2D problem. Decision boundaries of the models are presented when $N = 200$. (Right) Test accuracy averaged over five runs on the colorized MNIST.

# E. Experiments

We verify our theory by using the in-context domain adaptation abilities of Transformers using two synthetic problems. The ICL domain adapter is compared with 1. IWL with uLSIF using the RBF kernel; 2. DANN of a two-layer neural network with the ELU activation function; and 3. the same neural network trained on the source domain only. For the in-context learning, we used an eight-layer Transformer with eight heads and pre-trained it to minimize $\gamma(\mathrm{TF}_{\boldsymbol{\theta}}(\boldsymbol{x}'_*; \mathcal{D}'_S, \mathcal{D}'_T), y'_*)$, where $(\boldsymbol{x}'_*, y'_*) \sim \mathcal{D}'_T$, for randomly synthesized datasets $(\mathcal{D}'_S, \mathcal{D}'_T)(\neq (\mathcal{D}_T, \mathcal{D}_S))$ for $10^4$ iterations. Each dataset consists of $n = n' = N/2$ data points. Note that test data used to report test accuracy are unseen during pre-training.

Figure 1 (Left) presents test accuracy on the two-moon 2D problem, where the target distribution is a rotation of the source one. The decision boundaries are also presented, and we can observe that the Transformer learns a smoother boundary than others. Additionally, Figure 1 (Right) demonstrates the results of the colorized MNIST problem, where each dataset consists of images of two digits resized to $8 \times 8$ pixels, and their background colors alter based on the domains, as presented at the rightmost. On both problems, ICL consistently achieves much better performance than the baselines. These results indicate that Transformer implements adaptive domain adaptation algorithms to given datasets.

