# OpenReview forum: "Automatic Domain Adaptation by Transformers in In-Context Learning"
_ICML.cc/2024/Workshop/ICL — ICML 2024 Workshop ICL Poster_

### Official Review · Reviewer_A7Ch · 2024-06-04
**Transformers can approximate and utilize appropriate domain adaptation methods**

**Rating:** 2
**Fit:** 3
**Confidence:** 2

**Workshop Review:**

Overall, this is a good paper which makes progress in the understanding of how transformers are able to perform ICL. The main contributions are:
1. Proving that transformers can solve domain adaptation problems by approximating certain UDA algorithms.
2. Proving that transformers can automatically adapt to suitable domain adaptation algorithms based on the input dataset.

The biggest downside is the weak experiments. See comment below.

Suggestion: This should be an 8-page paper as the material in the Appendix is essential and not supplementary to understanding the paper (especially the experiments). Also, you could use the extra pages to space out all the equations which are difficult to parse inline.

Typos:
- In the Introduction UDA is used before defining it. It is instead defined in the next section.
- Last sentence on page 2 missing the word be before “the ReLU function”.
- At the top of page 3, “to indicate the number of layer” should read “to indicate the layer number”.
- The last sentence in Section 2.2: Transformer, starting with “We also…” does not make any sense. There is a word or two missing.
- The conclusion states that the experiments support the findings, but no experiments are introduced in the main paper – they are in the Appendix, which is not referenced.
- In Section 3.1, B is not defined and I am not sure what it is.

**Reason For Not Giving Higher Score:**

The experiments are weak - they are simply toy experiments. It would have been good to see if the results held up with larger and more realistic datasets such as Meta-Dataset [1] or VTAB [2].

[1] Triantafillou, Eleni, et al. "Meta-dataset: A dataset of datasets for learning to learn from few examples." arXiv preprint arXiv:1903.03096 (2019).

[2] Zhai, Xiaohua, et al. "A large-scale study of representation learning with the visual task adaptation benchmark." arXiv preprint arXiv:1910.04867 (2019).

**Reason For Not Giving Lower Score:**

The main contributions listed above justify the assigned score.

---

### Official Review · Reviewer_KZA1 · 2024-06-07
**This paper focuses on the unsupervised domain adaptation (UDA) problem and demonstrates the existence of transformer construction to address it.**

**Rating:** 2
**Fit:** 3
**Confidence:** 2

**Workshop Review:**

This paper explores the use of transformers in solving UDA by considering two types of algorithms: instance-based and feature-based methods. It proves that transformer architectures have the capacity to implement both algorithms effectively.

**Reason For Not Giving Higher Score:**

1. The paper lacks detailed construction methods for implementing the algorithms using transformers.
2. The experimental implementation details are insufficient. The results suggest that transformers can implement more advanced algorithms than IWL and DANN, but the paper does not discuss these findings in depth.

**Reason For Not Giving Lower Score:**

1. The paper is well-organized and clearly presented.
2. The study introduces a novel perspective on ICL by examining its application in unsupervised domain adaptation.

---

### Meta-Review · Area_Chair_B6Uh · 2024-06-14

**Recommendation:** 2

**Metareview:**

This paper demonstrates how Transformer models can address unsupervised domain adaptation (UDA) within the in-context learning framework by approximating both instance-based and feature-based UDA algorithms and automatically selecting the most suitable approximated algorithms for a given dataset.

All reviewers consider this paper favorably, but suggest expanding the experiment section.

---

### Decision · Program_Chairs · 2024-06-17

Accept (Poster)